# Clinical Presentation of Immune-Related Endocrine Adverse Events during Immune Checkpoint Inhibitor Treatment

**DOI:** 10.3390/cancers14112687

**Published:** 2022-05-29

**Authors:** Jenny Hui Ling Chieng, Zaw Win Htet, Joseph J. Zhao, E Shyong Tai, Sen Hee Tay, Yiqing Huang, Alvin Wong, Samantha Peiling Yang

**Affiliations:** 1Yong Loo Lin School of Medicine, National University of Singapore, 1E Kent Ridge Road, Singapore 119228, Singapore; josephjohnathan.zhao@mohh.com.sg (J.J.Z.); mdctes@nus.edu.sg (E.S.T.); 2Division of Endocrinology, Department of Medicine, National University Hospital, 5 Lower Kent Ridge Road, Singapore 119074, Singapore; zawwinhtet1991@gmail.com; 3Division of Rheumatology, Department of Medicine, National University Hospital, 5 Lower Kent Ridge Road, Singapore 119074, Singapore; sen_hee_tay@nuhs.edu.sg; 4Department of Haematology-Oncology, National University Cancer Institute, 5 Lower Kent Ridge Road, Singapore 119074, Singapore; yiqing_huang@nuhs.edu.sg (Y.H.); alvin_sc_wong@nuhs.edu.sg (A.W.)

**Keywords:** immune-related endocrine adverse events, immune checkpoint inhibitor treatment, thyroid dysfunction, pituitary dysfunction, insulin-dependent diabetes mellitus, survival analysis

## Abstract

**Simple Summary:**

With the growing use of immune checkpoint inhibitors (ICIs), clinicians are increasingly encountering a unique set of adverse events called immune-related adverse events (irAEs) associated with ICIs. In this retrospective study, we tried to analyse the clinical course of endocrine irAEs and determine the factors associated with persistent endocrine dysfunction. We also performed survival analysis of patients with endocrine irAEs compared to those without any irAEs. The clinical course of endocrine irAEs was found to be highly variable. The presentation of some endocrine irAEs could be nonspecific, and sometimes life-threatening if not detected early, e.g., hypocortisolism and diabetic ketoacidosis. Therefore, frequent clinical assessment and laboratory monitoring are recommended. Measuring thyroid antibodies at the start of thyroid dysfunction could be helpful as it was found to be associated with persistent thyroid dysfunction in our study. The presence of endocrine irAEs was found to have survival benefits in our patient population.

**Abstract:**

The exact clinical course and factors associated with persistent endocrine immune-related adverse events (irAEs) are not well-established. Elucidation of these information will aid irAEs screening and follow-up planning for patients on immunotherapy. We analysed the clinical course of endocrine irAEs including thyroid and pituitary dysfunction and insulin-dependent diabetes mellitus (IDDM), identified factors associated with persistent thyroid dysfunction, and determined the association between endocrine irAEs and survival parameters. This retrospective observational study enrolled patients with metastatic cancer who underwent anti-PD-1, anti-PD-L1, and/or anti-CTLA-4 treatment and developed endocrine irAE at the National University Cancer Institute, Singapore, between June 2015 and December 2020. Sixty-six patients with endocrine irAE were evaluated, with a median follow-up time of 15.7 months. The median time to onset of thyroid dysfunction, pituitary dysfunction, and IDDM was 1.8 months (range: 0.3–15.8 months), 6.8 months (range: 1.5–27.3 months), and 7.8 months (range: 1.4–9.1 months), respectively. Positive thyroperoxidase antibodies (TPOAb) and/ or thyroglobulin antibodies (TgAb) status at the time of thyroid dysfunction was associated with persistent thyroid dysfunction (OR 11.6, 95% CI 1.3–570.8, *p* = 0.02; OR 8.8, 95% CI 1.3–106.9, *p* = 0.01, respectively). All patients with pituitary irAE had central hypocortisolism. All patients with IDDM had grade 4 irAE. Patients with endocrine irAE had longer median survival times. Endocrine irAEs were associated with non-progressive disease. The screening and follow-up approach for endocrine irAEs should be tailored according to each endocrinopathy’s clinical course. Early screening is imperative given its wide median time to onset.

## 1. Introduction

The development of immune checkpoint inhibitors (ICIs) has revolutionised cancer treatment, remarkably improving the clinical outcomes for a variety of malignancies [1,2]. While ICIs have a unique toxicity profile compared to other systemic anticancer agents, they have led to the advent of a new spectrum of adverse events termed immune-related adverse events (irAE) [3,4,5], which can occasionally be fatal [6]. irAEs are defined as side effects that occur after initiation of ICI treatment with an underlying immunological basis [5,7,8]. These irAEs can affect any organ, though most commonly the skin, gastrointestinal tract, endocrine organs, and lungs are affected [9,10]. In a recent study at our centre, endocrinopathies were found to be the most common irAEs (8.1%) [11]. In a meta-analysis and systematic review, incidence of endocrine irAE have been reported as 6.6% for hypothyroidism, 2.9% for hyperthyroidism, 1.3% for hypophysitis, and 0.2% for insulin-dependent diabetes mellitus (IDDM) [12]. The exact timeline and predictors of persistent hormonal dysfunction in endocrine irAEs, including thyroid and pituitary dysfunction, and IDDM, have not been well-established yet. These features have direct implications on the frequency of screening and follow-up for endocrine irAEs in patients on ICIs and those who develop endocrine irAE subsequently.

As recent studies have suggested that the presence of irAE is associated with the survival outcomes [13,14,15,16,17,18,19,20], those with irAEs being associated with a better prognosis compared to those without irAEs [21,22], and the same association being observed for endocrine irAEs [23], this study also seeks to validate these findings by assessing the implication of endocrine irAEs on survival outcomes.

In our study, we aimed to evaluate the clinical course of endocrine irAEs, identify the factors associated with persistent dysfunction, and assess the prognostic significance of endocrine irAEs.

## 2. Methods

### 2.1. Patient Eligibility and Study Design

This was a retrospective observational study in a single tertiary referral centre that enrolled patients with confirmed diagnosis of metastatic cancer who underwent treatment with anti-CTLA-4 (ipilimumab or tremelimumab), anti-PD-1 (nivolumab, zimberelimab, pembrolizumab, or spartalizumab), anti-PD-L1 (durvalumab, atezolizumab, or FAZ053), or combinations of these drugs, and developed endocrine irAE at the National University Cancer Institute, Singapore, between June 2015 and December 2020. The study protocol was approved by the local ethics board committee (NHG DSRB Study Reference Number: 2021/00631).

Primary outcomes included characterisation of the clinical course of endocrine irAE and identification of the factors associated with persistent dysfunction. A secondary outcome was the evaluation of association between endocrine irAEs and survival parameters.

Baseline demographic, treatment characteristics, laboratory tests, and imaging data were curated from the Computerized Patient Support System (CPSS) medical record database. The primary malignancies were grouped into the following categories: head and neck (nasopharyngeal carcinoma, oral cavity carcinoma, and oropharyngeal carcinoma), lung, gastrointestinal (oesophageal carcinoma, gastric carcinoma, colorectal carcinoma, and hepatocellular carcinoma), renal, skin (melanoma), gynaecological (ovarian carcinoma, vulva carcinoma, and endometrial carcinoma), and others.

Clinical severity of the irAEs was graded according to the US National Cancer Institute Common Toxicity Criteria for Adverse Events (NCI-CTCAE; version 5.0), while treatment efficacy was evaluated with Response Evaluation Criteria in Solid Tumours (RECIST) criteria (version 1.1), and survival data of patients were collected for all types of endocrine irAE.

For surveillance of irAE, thyroid function test was screened at baseline, then every 4 to 6 weeks from immunotherapy initiation, or when symptomatic. In the event of thyrotoxicosis, thyroid hormone profile was assessed every 2 to 3 weeks to detect transition to hypothyroidism. For thyroid irAE, subclinical hypothyroidism was defined as normal free thyroxine (FT4) and increased thyroid stimulating hormone (TSH) levels, subclinical thyrotoxicosis as normal FT4 and decreased TSH levels, overt primary hypothyroidism as decreased FT4 and increased TSH levels, and overt thyrotoxicosis as increased serum FT4 and decreased TSH levels. Those who required anti-thyroid medications or supplementation with levothyroxine were followed up with thyroid function tests every 4 to 6 weeks and decision to adjust or stop the medications were made based on thyroid function and clinical assessment by the treating clinician. Those requiring anti-thyroid medications or thyroid hormone supplementation at the point of the study end-date were classified as having persistent thyroid irAE, while the converse was considered as having resolved thyroid irAE. Similarly, for pituitary irAE and IDDM irAE, the requirement of steroid or levothyroxine replacement for hypopituitarism or insulin for IDDM, respectively, was considered as having persistent pituitary or IDDM irAE.

Serum sodium and potassium were monitored regularly at least every 4 to 6 weeks. Thyroid autoimmune antibodies, pituitary hormonal tests, serum glucose, HbA1c, glutamic acid decarboxylase (GAD) antibodies, anti-islet cell antibodies, and imaging were assessed according to clinical indications.

### 2.2. Laboratory Assays

Free thyroxine (fT4), thyroid stimulating hormone (TSH), serum cortisol, follicle stimulating hormone (FSH), luteinising hormone (LH), testosterone, oestradiol, and prolactin were measured using chemiluminescent immunoassays (Beckman DXI, Brea, CA, USA). Plasma adrenocorticotrophic hormone (ACTH), serum growth hormone (GH), and insulin-like growth factor-1 (IGF-1) were measured using chemiluminescent immoumetric assay (Siemens IMMULITE, Erlangen, Germany). HbA1C was measured using high-performance liquid chromatography (Bio-Rad D100/TURBO V2.0, South Granville, NSW, Australia). Serum C-peptide was measured using chemiluminescent immunoassay (ADVIA Centaur, Siemens Healthcare, Erlangen, Germany). TSH receptor antibodies (TRAb) were measured by enzyme immunoassay (ELISA). Thyroperoxidase antibodies (TPOAb) and anti-GAD were measured using enzyme-linked immunosorbent assay (ELISA). Thyroglobulin antibodies (TgAb) were measured using immunoassays (QL immulite, Siemens Healthcare, Erlangen, Germany), Immulite 2000 (Siemens), Kryptor (Brahms, Hennigsdorf, Germany), e411 (Roche, Basel, Switzerland), or e801 (Cobas, Roche Diagnostics, Basel, Switzerland).

### 2.3. Statistical Analyses

Data were represented as means and standard deviations, medians, and their ranges, as well as frequencies and percentages where appropriate. Time-varying incidences of all-grade irAE after commencement of ICI were examined using the Royston–Parmar spline model. The 95% CI of incidence rates were estimated with a Poisson distribution. This was performed using R-4.0.0 (with packages ‘rstpm2′ and ‘fmsb’, R Foundation for Statistical Computing, Vienna, Austria). Associations between clinical parameters and the development of persistent thyroid dysfunction were examined using Fisher’s exact test. Survival analysis was performed using a separate study population that enrolled 142 patients with confirmed diagnoses of cancer who underwent treatment with anti-PD-1/PD-L1 and/or anti-CTLA-4 without developing any irAE as the comparison group. Overall survival (OS), defined as the time from the initiation of immunotherapy to occurrence of death, was depicted using the Kaplan–Meier method. Comparison between survival curves were assessed using log-rank test. The Cox-regression model was used to generate hazard ratios. Patients who were still alive at the point of the study end date were censored. For all statistical tests, *p*-value < 0.05 was considered statistically significant. These were computed using IBM SPSS Statistics Version 26.

## 3. Results

### 3.1. Overall Study Cohort

#### 3.1.1. Patient and Treatment Characteristics

Sixty-six cancer patients with endocrine irAE were evaluated. Patient and treatment characteristics are summarised in Table 1. The mean age was 60.5 years (standard deviation: 10.6 years); 57.6% (38/66) were males. The most common primary malignancy in this cohort was lung cancer (33.3%; 22/66); 84.8% (56/66) of patients received anti-PD-1/PD-L1 treatment only, 3% (2/66) received anti-CTLA-4 treatment only, and 12.1% (8/66) received both. The majority [93.9% (62/66)] of patients had a single endocrinopathy, of which 72.6% (45/62) had thyroid dysfunction, 24.2% (15/62) had pituitary dysfunction, and 3.2% (3/62) had IDDM. All four patients with multiple endocrinopathies were on anti-PD-L1 therapy. The median follow-up time was 15.7 months (range: 1.8–60.5 months).

#### 3.1.2. Incidence of Endocrine irAE

The time-varying incidence of thyroid and pituitary irAE are shown in Figure 1. For thyroid irAE, patients either developed biphasic thyroiditis with an initial hyperthyroidism phase followed by hypothyroidism, hyperthyroidism only, or hypothyroidism only. The peak incidence of patients of the initial hyperthyroid phase in those with biphasic thyroiditis was 3.81 weeks. The peak incidence rate of subsequent conversion to hypothyroidism was at 9.05 weeks. The peak incidence of those who only developed hyperthyroidism or hypothyroidism was 3.06 weeks and 15.76 weeks, respectively. The peak incidence of patients who developed pituitary dysfunction was at 7.6 weeks.

### 3.2. Thyroid irAE Analysis Characterisation

#### 3.2.1. Patient and Treatment Parameters

Forty-nine (74.2%) cancer patients had thyroid irAE; 24.5% (12/49) of patients had pre-existing thyroid dysfunction, of which 50% (6/12) had subclinical hypothyroidism, 33.3% (4/12) had subclinical hyperthyroidism, and 14.3% (2/12) had Graves’ disease in remission. The median time to onset of thyroid dysfunction after immunotherapy was started was 1.8 months (range: 0.3–15.8 months). Most (67.4% (33/49)) of these patients were asymptomatic, with deranged thyroid hormones detected during routine toxicity screening; 71.4% (35/49) of patients required levothyroxine eventually. The median dosage of levothyroxine required during thyroid dysfunction was 1.25 mcg/kg/day (range: 0.3–3.5 mcg/kg/day). All the thyroid irAEs were mild; 81.6% (40/49) of patients developed grade 2 irAE and 18.4% (9/49) developed grade 1 irAE. The characteristics of patients with thyroid irAEs are summarised in Table 2.

Among the patients who developed thyroid irAE and had PET-CT scans for surveillance of underlying malignancy after initiation of ICI therapy, physiological FDG uptake was seen in 72.7% (8/11) of patients and the remaining 27.3% (3/11) had thyroid abnormalities in the PET-CT scan. These abnormalities included intense FDG uptake involving both thyroid lobes in one case and an atrophic thyroid gland in two cases. The patient with intense FDG uptake had positive TPOAb and TgAb, with negative TRAb, and was biochemically hypothyroid at the time of the PET-CT scan. This patient was not known to have any pre-existing thyroid disease; baseline thyroid function tests were normal and the baseline PET-CT scan also had no significant thyroid findings. The increased FDG uptake with the associated biochemical picture after initiation of ICI suggests ICI-associated thyroiditis. Both patients with atrophic thyroid glands had normal-looking thyroid glands in baseline CT/PET-CT prior to the initiation of ICIs.

#### 3.2.2. Clinical Course of Thyroid Dysfunction

The evolution of thyroid dysfunction over time is summarised in Figure 2. At diagnosis of thyroid dysfunction, 89.8% (44/49) of patients had overt thyroid dysfunction, of which 25% (11/44) had hypothyroidism and 75% (33/44) had thyrotoxicosis. The hypothyroidism was persistent in 91% (10/11) of patients. The thyrotoxicosis was persistent in 9.1% (3/33) of patients, all of whom had positive TRAb. The first patient had pre-existing Graves’ disease and had a relapse of Graves’ disease after initiation of ICI therapy with elevated fT4 of 17.6 pmol/L and suppressed TSH of 0.02 mIU/L. The second patient had no pre-existing thyroid disease and developed subclinical thyrotoxicosis with normal fT4 of 11.5 pmol/L, and suppressed TSH of 0.28 mIU/L, not requiring anti-thyroid medications. The last patient had no pre-existing thyroid disease and developed overt thyrotoxicosis with elevated fT4 of 45.3 pmol/L and suppressed TSH of 0.01 mIU/L, requiring an anti-thyroid drug. This patient subsequently died of the underlying cancer within a short period of follow-up (2.3 months) after the diagnosis of thyrotoxicosis. In 87.9% (29/33) of patients with the initial thyrotoxic phase, hypothyroidism developed subsequently, of which the hypothyroidism was persistent in 79.3% (23/29) of patients. As for the minority of patients who developed subclinical thyroid dysfunction (10.2%; 5/49), 40% (2/5) had subclinical hypothyroidism that was persistent, requiring thyroid hormone replacement. The first patient had positive TPOAb and TgAb with negative TRAb at the time of subclinical hypothyroidism, while the other patient did not have any thyroid antibody evaluation. In total, 60% (3/5) had subclinical thyrotoxicosis that resolved subsequently without a hypothyroid phase. All these three patients had negative TPOAb, TgAb, and TRAb at the time of subclinical hyperthyroidism.

#### 3.2.3. Predictors of Persistent Thyroid Dysfunction

The following factors were analysed for association with persistent thyroid dysfunction: thyroid autoantibody status, type of immunotherapy administered, and thyroid uptake of FDG-PET after initiation of ICI therapy.

The prevalence of thyroid autoantibodies (if available) at the time of thyroid dysfunction showed that 50% (21/42) of patients had positive TPOAb, 60% (21/35) of patients had positive TgAb, and 6.1% (3/22) of patients had positive TRAb.

There was an association between TPOAb positivity and persistent thyroid dysfunction, where the odds ratio for persistent thyroid dysfunction among patients with positive TPOAb compared to patients with negative TPOAb was 11.6 (95% confidence interval: 1.3 to 570.8; *p*-value: 0.02). Only one patient with positive TPOAb had resolved irAE. Amongst the patients with positive TPOAb with persistent thyroid dysfunction, 95% (19/20) of patients developed persistent hypothyroidism requiring long-term thyroxine replacement. The last patient had Graves’ disease with both positive TPOAb and TRAb, resulting in persistent thyrotoxicosis. Amongst 21 patients with negative TPOAb, 70% (7/10) of those with persistent thyroid dysfunction had positive TgAb, while only 33% (2/6) of patients with resolved irAE had positive TgAb.

TgAb positivity was associated with persistent thyroid dysfunction, where the odds ratio for persistent thyroid dysfunction among patients with positive TgAb compared to patients with negative TgAb was 8.8 (95% confidence interval: 1.3 to 106.9; *p*-value: 0.01). Amongst the patients with positive TgAb, 90.5% (19/21) developed persistent hypothyroidism requiring long-term thyroxine replacement. All patients who had positive TRAb had persistent thyrotoxicosis (3/22) during our study.

All patients who were prescribed a combination of anti-PD-1/anti-PD-L1 and anti-CTLA-4 (6/49) and developed thyroid irAEs had persistent thyroid dysfunction during our study period.

The only patient with intense thyroid FDG uptake on PET-CT scan and two patients with atrophic thyroid gland on PET-CT scan had persistent thyroid dysfunction during our study period, whereas, among those with physiological thyroid FDG uptake on PET-CT scan, 75% (6/8) had persistent thyroid dysfunction and 25% (2/8) had resolved irAE.

#### 3.2.4. Survival Analysis and Treatment Response

The Kaplan–Meier curves for the OS of the patients who developed thyroid irAE and the comparison study population of patients without irAE are shown in Figure 3. The presence of thyroid irAE was associated with longer median OS (39 weeks vs. 11.6 weeks; *p*-value < 0.001).

### 3.3. Pituitary irAE Analysis

#### 3.3.1. Patient and Treatment Characteristics

Eighteen patients had pituitary irAE; 94.4% (17/18) of patients had central hypocortisolism only. Only one patient had multiple pituitary hormone deficiencies (central hypocortisolism and central hypogonadism) and this patient also had thyroid irAE with a biphasic thyroiditis response and persistent overt primary hypothyroidism with positive TPOAb and TgAb status. The patient received nivolumab, PD-1 ICI therapy, and developed thyroid irAE 3 months later, and pituitary irAE after 10 months of PD-1 therapy. The median time to the onset of pituitary dysfunction after immunotherapy was started was 6.8 months (range: 1.5–27.3 months). In the sub-set of patients who received anti-PD-1/anti-PD-L1 treatment only, the median time to the onset of pituitary dysfunction was 6 months (range: 1.5–27.3 months). In the sub-set of patients who received anti-CTLA-4 treatment only, the median time to the onset of pituitary dysfunction was earlier at 5 months (range: 3–6.9 months); 72.2% (13/18) of patients were symptomatic at presentation. The most frequently reported symptoms included postural giddiness, fatigue, and loss of appetite. In terms of severity of irAE, 83.3% (15/18) of patients developed grade 2 irAE and 16.7% (3/18) developed grade 3 irAE; 72.2% (13/18) of patients had MRI pituitary scans done. Amongst the patients who underwent an MRI scan, 23.1% (3/13) of patients had incidental findings of hypo-enhancing sub-centimetre lesions in the anterior pituitary gland, likely to represent incidental pituitary microadenoma or Rathke’s cleft cyst. However, they did not have the typical findings of hypophysitis such as diffuse enlargement of the pituitary gland and thickening of the pituitary stalk, and no patient had signs and symptoms suggestive of increased intracranial pressure or optic nerve compression. The characteristics of patients with pituitary irAEs are summarised in Table 3.

#### 3.3.2. Survival Analysis and Treatment Response

The Kaplan–Meier curves for OS for the patients who developed pituitary irAE and the control study population of patients without irAE are shown in Figure 4. The presence of pituitary irAE was associated with longer median OS (38.9 weeks vs. 11.6 weeks; *p*-value < 0.001).

### 3.4. Insulin-Dependent Diabetes Mellitus irAE Analysis

Three patients had insulin-dependent diabetes mellitus (IDDM) irAE. All three patients were not known to have any pre-existing diabetes mellitus. The median time to the onset of IDDM after immunotherapy was started was 7.8 months (range: 1.4–9.1 months). At the onset of IDDM, median HbA1c was 9.5% (range: 7.8–9.8). The pancreatic beta-cell reserve was assessed in one of these patients. This patient had a low fasting C-peptide level of 131 pmol/L (normal: 364–1655 pmol/L) with a paired plasma glucose of 24.5 mmol/L (normal: 4–7.8 mmol/L). Anti-GAD antibody was negative in all three patients. Anti-islet cell antibody was negative in one patient and was not evaluated in the rest. All three patients presented with diabetic ketoacidosis (DKA) after the initiation of ICIs and had grade 4 irAE.

### 3.5. Survival Analysis and Treatment Response in Patients with Endocrine irAEs

There was a significant difference in survival times between patients with any form of endocrine irAE and those without irAE (log-rank test *p*-value < 0.001). The estimated median survival times were 39 weeks in patients with endocrine irAE and 11.3 weeks in patients without irAE. The hazard ratio for death in patients with endocrine irAE was 0.34 (95% CI: 0.22–0.53, *p* < 0.001).

When separate analysis was done on patients with lung cancer, there was also a significant difference in survival times between patients with endocrine irAE and patients without irAE (log-rank test *p*-value = 0.001). Among patients with lung cancer, the estimated median survival times were 33.5 weeks in patients with endocrine irAE and 14 weeks in patients without irAE. The hazard ratio in patients with endocrine irAE was 0.0244 (95% CI: 0.098–0.607, *p* = 0.002). Similarly, among patients with gastrointestinal malignancies, there was a significant difference in survival times between patients with endocrine irAE and patients without irAE (log-rank test *p*-value = 0.031). The estimated median survival times were 25.7 weeks in patients with endocrine irAE and 8.7 weeks in patients without irAE with the hazard ratio for death in patients with endocrine irAE 0.437 (95% CI: 0.202–0.946, *p* = 0.036). Among patients with renal cell carcinoma, there was no significant difference in survival times between patients with endocrine irAE and patients without irAE (log-rank test *p*-value = 0.17). The hazard ratio in patients with endocrine irAE among patients with renal cell carcinoma was 0.442 (95% CI: 0.133–1.472, *p* = 0.184). The Kaplan–Meier curves for the OS for patients with endocrine irAEs and those without irAEs are shown in Figure 5.

The presence of endocrine irAE was associated with nonprogressive disease. The odds ratio for non-progressive disease in patients with endocrine irAE compared to patients without irAE was 173.1 (95% confidence interval: 27.0 to 6962.0; *p*-value < 0.001). Evaluation of the latest treatment response of patients who developed endocrine irAE showed that 40.9% (27/66) of patients had progressive disease, 48.5% (32/66) of patients had stable disease, and 10.6% (7/66) of patients had partial response. There were no patients with a complete response. The difference in RECIST status remained significant when separate analyses were done for thyroid and pituitary endocrine irAEs.

## 4. Discussion

We found that endocrine irAEs could arise between 0.3 to 27.3 months. Most of thyroid irAEs and all the pituitary and IDDM irAEs were persistent. The presence of thyroid autoantibody during thyroid irAE occurrence (TPOAb and/or TgAb) were found to be factors associated with persistent thyroid dysfunction. Patients who developed endocrine irAEs had longer median OS compared to those who did not develop any irAEs, suggesting that the development of endocrine irAEs confers survival benefits in patients on immunotherapy.

Our study has some important limitations. First, it is a retrospective study. Prospective studies with serial monitoring of biochemical markers, flow cytometry, and cytokines can better improve our understanding of the mechanisms of the irAEs. Second, our sample size is relatively small. For example, there were only three patients with IDDM in the study, thereby making it difficult to draw conclusions on the survival benefits and clinical course definitively. Furthermore, it contained a small number of cases with a variety of cancers. A larger number of cases with different types of cancer cases is needed to be able to detect if there are differences in impact of endocrine irAEs on individual types of cancer. In addition, it is a single-centre study, and therefore the findings may not be representative of the population around the globe.

### 4.1. Clinical Course of Endocrine irAEs and Monitoring

Overall, the time to onset of endocrine irAEs after ICI initiation had a wide range (0.3–27.3 months), indicating the need for early screening of endocrine irAEs after starting ICIs and continued surveillance during the treatment course.

As most patients with thyroid irAEs were asymptomatic, screening for thyroid irAEs should be performed, in addition to clinical assessment. In our study, the median time to the onset of thyroid dysfunction was 1.8 months (range: 0.3–15.8 months), which is similar to that of other studies [24,25,26,27,28]. An initial thyroid function test should thus be performed early after ICI initiation to allow for prompt detection of thyroid irAEs. This is in line with the routine 4- to 6-week screening intervals recommended by the ESMO, ASCO, NCCN, and SITC Clinical Practice Guidelines [9,10,29,30]. For patients who developed thyroid irAE, our study found that most patients (59.2%; 29/49) had a biphasic thyroiditis response with an initial phase of thyrotoxicosis (not requiring anti-thyroid medications), which was followed by persistent hypothyroidism in the majority of patients (79.3%; 23/29). This is similar to a recent study that showed that amongst those who developed thyroid irAE post-ICI therapy, the majority developed a biphasic thyroiditis response. [27,28] The peak incidence rates of conversion to hypothyroidism in patients with biphasic thyroiditis was 9.1 weeks, hence, it is recommended that patients with thyrotoxicosis should be followed-up with a thyroid function test every 4–6 weeks to ensure that the transition to the hypothyroidism phase is detected early.

The median time to onset of pituitary irAEs in our study was 6.8 months (range: 1.5–27.3 months). While most of these patients (72.2%; 13/18) were symptomatic, clinical assessment for pituitary irAE can still be challenging for clinicians because of its varied manifestations that are sometimes difficult to differentiate from other acute illnesses and the underlying malignancy. Similar to lymphocytic hypophysitis, corticotroph- and thyrotroph-secreting cells are preferentially affected in immunotherapy-related hypophysitis, though diabetes insipidus is less commonly reported, unlike in lymphocytic hypophysitis [31,32,33,34,35,36]. Central hypocortisolism had been reported to be persistent in most patients with immunotherapy-related hypophysitis [31,32,33,34,35,36]. Similar to literature, all our study patients with pituitary irAE had central hypocortisolism, while one patient also had multiple pituitary hormone deficiencies. The most frequently reported symptoms include postural giddiness, fatigue, and loss of appetite. Other common manifestations of ICI-induced pituitary dysfunction reported in other studies include headache and visual changes [37,38]. Clinicians should be mindful of these symptoms that may suggest the development of pituitary irAE. Diagnosis can subsequently be confirmed in patients with suggestive symptoms via biochemical tests. The ASCO Clinical Practice Guidelines recommend the diagnostic workup to include baseline investigations such as 8 am cortisol, ACTH, thyroid function test, and electrolytes. Stimulation tests like the 250 mcg cosyntropin stimulation test can also be obtained [38]. Other tests such as for LH, FSH, testosterone (in males), and estrogen (in pre-menopausal females) can be considered in patients with symptoms suggestive of hypogonadism, and brain MRI can be considered in patients with two or more pituitary hormonal axes involvement [10] or with symptoms of raised intracranial pressure. In comparison, the ESMO Clinical Practice Guidelines recommend MRI for all patients [9]. In our study, brain MRI was performed for 13 patients, all of whom had central hypocortisolism. All 13 scans did not show the typical features of hypophysitis, reflecting the lack of utility of such scans in patients in the absence of multiple pituitary hormonal axes involvement. This supports the limitation of use of MRI brain scans in patients with involvement of two or more pituitary axes as per the ASCO guidelines. The onset of pituitary irAE in our study was generally later than that described in the literature. It could be related to the fact that the majority of our pituitary irAE patients were on anti-PD-1/anti-PD-L1 therapy rather than anti-CTLA-4 therapy, which is generally associated with hypophysitis. This can be further validated in a larger study.

A total of three patients with IDDM were identified in our study. The time to onset of IDDM ranged from 1.4–9.1 months. None of these patients underwent regular blood glucose measurements for screening of diabetes mellitus after initiation of ICIs. All of them presented with DKA and were subsequently diagnosed with IDDM. Given the life-threatening nature of DKA, it is imperative that clinicians have an effective screening strategy to detect cases of IDDM during ICI therapy. The ASCO guidelines propose screening via symptoms and signs of hyperglycaemia, as well as measurement of blood glucose levels at baseline and with each treatment cycle [10]. In our study, patients with IDDM had elevated HbA1c levels in the range of 7.8–9.8% when they presented with DKA. This suggests that the onset of hyperglycaemia in our study population was at least a few weeks prior to the DKA presentation, and these patients could benefit from earlier blood glucose screenings. In comparison, other studies report a lower HbA1c of 7.6% at presentation [39,40], possibly due to earlier detection of IDDM. However, the onset of the DM post-ICI therapy can also be fulminant, as described in a systematic review [40]. Thus, routine blood glucose monitoring may not always be beneficial.

The variable presentation and clinical course of endocrine irAEs as seen in our study highlight the need to develop better strategies to mitigate the effects of irAEs. As outlined in a recent article by Aung Naing et al., strategies such as providing better patient education, harnessing technology for early recognition and prompt intervention of irAEs, and conducting further studies to improve our understanding of irAEs and to develop personalized irAE management are important to improve our management of irAEs [41].

### 4.2. Prognostic Significance of Endocrine irAE

The presence of irAEs has been associated with better survival outcomes including longer OS [42,43,44]. We showed that patients who developed endocrine irAEs had longer median OS compared to those who did not develop any irAEs. This was also observed in the subgroup analysis of patients who developed thyroid or pituitary irAEs, suggesting that the development of endocrine irAEs may confer survival benefits. The underlying mechanism behind this is hypothesized to be due to common antigens shared by tumours and target organs affected by irAEs, with the development of irAEs reflecting the anti-tumour response [16,45]. Nevertheless, these results should be interpreted with caution because of the lack of adjustment for confounders such as individual cancer type and stage in this present analysis, which can have a significant impact on the prognosis of patients. In addition, the possibility of lead-time bias should also be considered, where only those with increased survival from ICI therapy are followed up long enough to develop irAEs [46]. A recent study had showed that longer OS and progression-free survival was observed only in patients with overt thyrotoxicosis from thyroid irAEs, but not in those with overt hypothyroidism, as compared to patients without thyroid irAEs, after adjusting for age, gender, brain metastases, and ICI-type [28].

### 4.3. Factors Associated with Persistent Thyroid irAEs

Several studies had shown that the presence of positive TPOAb and TgAb at baseline was significantly associated with destructive thyroiditis in patients treated with ICI [28,47]. In our study, the presence of TPOAb and TgAb during ICI treatment was also found to be significantly associated with persistent thyroid dysfunction. Most of these patients developed persistent hypothyroidism requiring long-term thyroxine replacement. The presence of TPOAb and TgAb during ICI-induced thyroid irAE can be used as a predictor for the persistence of the thyroid dysfunction and the subsequent need for long-term thyroxine requirement. Similar to other studies [27], the median dosage of levothyroxine replacement in our study was 1.25 mcg/kg/day. This is lower than the usual dose required in post-thyroidectomy patients [48,49], suggesting the incomplete destruction of the gland in ICI-induced thyroiditis. For these patients, the ESMO guidelines recommend a thyroxine dose replacement of 0.5–1.5 mcg/kg/day, while the ASCO guidelines recommend an ideal-body-weight-based dose of approximately 1.6 mcg/kg/kg/day in those without risk factors, and the SITC guidelines recommend a dose of 1.5–1.6 mcg/kg in young, healthy patients.

The exact mechanism of the association between thyroid autoantibodies and thyroid irAEs is not known. The thyroid autoantibodies could be responsible for thyroid dysfunction after ICI therapy. On the other hand, the thyroid autoantibodies might also be the result of an immune response to the release of thyroid antigens during the destructive thyroiditis process. A previous study found that the thyroid autoantibodies developed shortly after commencement of pembrolizumab and hypothesised that ICIs uncovered underlying autoimmunity by modulating the auto-immune equilibrium [25]. Meanwhile, another study has found that patients with pembrolizumab-induced thyroiditis had elevated circulating CD56^+^CD16^+^ natural killer cells, without elevation of the CD3^+^ T-cell count, or the CD4^+^ and CD8^+^ T-cell sub-populations. These patients also had increased HLA-DR surface expression in CD14^+^CD16^+^ intermediate (pro-inflammatory) monocytes. The authors indicated that the mechanism of thyroid destruction might be T-cell, NK-cell, and/or monocyte mediated, rather than B-cell mediated [50].

The FDG avidity of the thyroid gland on PET-CT scan prior to ICI had been described to predict the development of overt thyroid irAEs after nivolumab therapy [51]. Our study analysed the PET-CT scans of patients with thyroid irAEs to determine if the FDG uptake of the thyroid gland was associated with persistent thyroid dysfunction after ICI therapy. Among the 11 patients who underwent PET-CT scans during ICI, one patient had bilateral intense FDG uptake in the thyroid gland, and two patients had atrophic thyroid glands. All these three patients developed persistent thyroid dysfunction. The patient with increased FDG uptake in the thyroid gland had a normal baseline thyroid FDG avidity before ICI initiation. After ICI commencement, this patient developed thyroiditis with a biphasic response together with new-onset increased FDG uptake in the thyroid gland during the hypothyroid phase. Given the temporal sequence of events, it is plausible that the ICI therapy had triggered thyroiditis in this patient, and the increased thyroid FDG uptake was related to the thyroiditis. The other two patients with atrophic thyroid glands could represent the “burnt-out” phase of the thyroiditis, given that both cases eventually developed severe hypothyroidism. Our data suggest that the thyroid function of patients with abnormal PET-CT scans while on ICI should be monitored long-term for the persistence of ICI-induced thyroid dysfunction. We acknowledge that the number of patients who underwent PET-CT scans during ICI is small in our study, and this needs to be validated in future studies.

Our study also analysed if the type of immunotherapy was associated with persistence of thyroid dysfunction. There was, however, no patient on monotherapy with anti-CTLA4 who developed thyroid dysfunction and thus analysis could only be performed for those on combination therapy versus those on anti-PD-1/anti-PD-L1 therapy. All patients on combination therapy had persistent thyroid dysfunction. However, an association between the type of immunotherapy and persistence of thyroid dysfunction would need to be evaluated in larger-scale studies.

### 4.4. Hypophysitis and CTLA-4 Therapy

A meta-analysis reported that the incidence of hypophysitis was higher with anti-CTLA-4 as compared to anti-PD-1 therapy [12], which may be related to pituitary expression of CTLA-4 [52]. In our study, there were only two patients on anti-CTLA-4 monotherapy and both developed pituitary irAEs. This is in line with the reported literature.

## 5. Conclusions

In conclusion, our study has shown that the time to the onset of endocrine irAEs after ICI initiation has a wide range; thus, early screening and continued surveillance of endocrine irAEs is imperative. To monitor for thyroid irAEs, an initial thyroid function test is recommended to be performed at the start of ICI therapy and repeated every 4–6 weeks. Additionally, it has been found that in those who develop thyroid irAEs, the presence of TPOAb and TgAb is associated with the persistence of thyroid dysfunction. For pituitary irAEs, clinicians should be vigilant of the nonspecific nature of pituitary irAEs. Laboratory investigations should be ordered promptly when suspicious symptoms are present. However, MRI can be limited to those with multiple pituitary hormonal axes derangement or symptoms of raised ICP. As patients with IDDM all presented with DKA in our study, it is important for clinicians to perform routine blood glucose monitoring in patients undergoing ICI treatment as well. Finally, the presence of thyroid and pituitary irAEs has been found to confer survival benefits in our study population. Being a single-centre retrospective study with a small number of cases, our findings should be confirmed in larger studies.

## Figures and Tables

**Figure 1 cancers-14-02687-f001:**
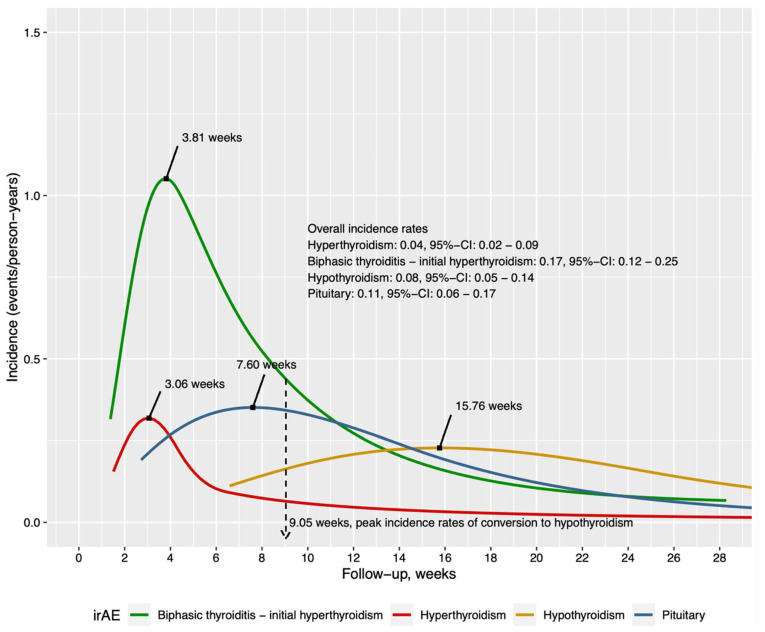
Time-varying incidence of respective irAE.

**Figure 2 cancers-14-02687-f002:**
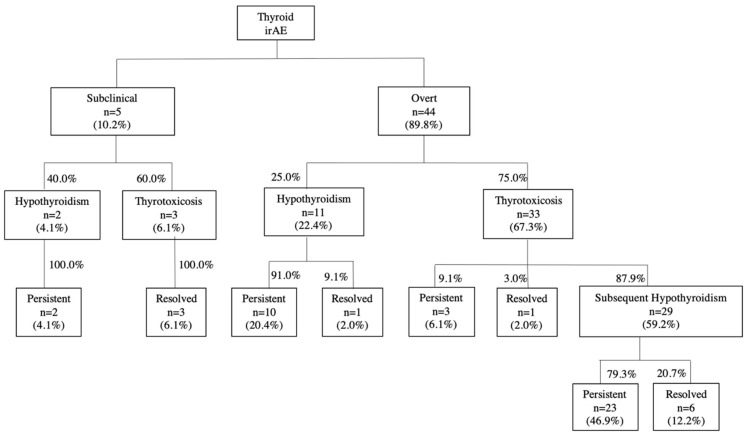
Evolution of thyroid dysfunction over time (*n* = 49).

**Figure 3 cancers-14-02687-f003:**
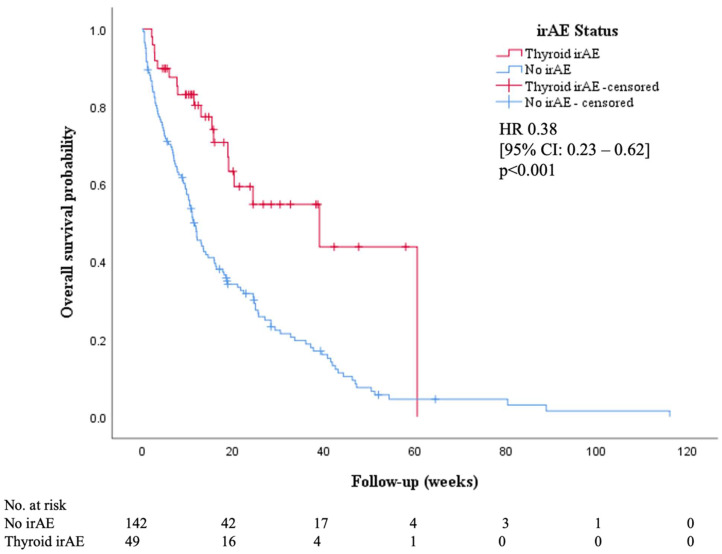
Kaplan–Meier curves for overall survival according to thyroid irAE status.

**Figure 4 cancers-14-02687-f004:**
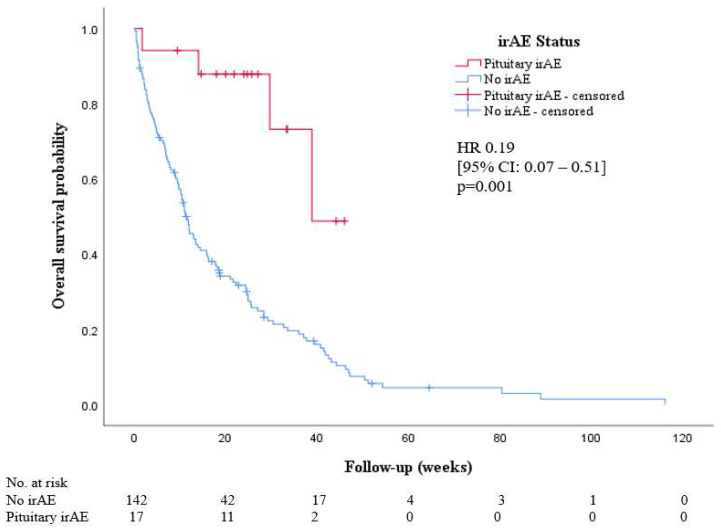
Kaplan–Meier curves for overall survival according to pituitary irAE status.

**Figure 5 cancers-14-02687-f005:**
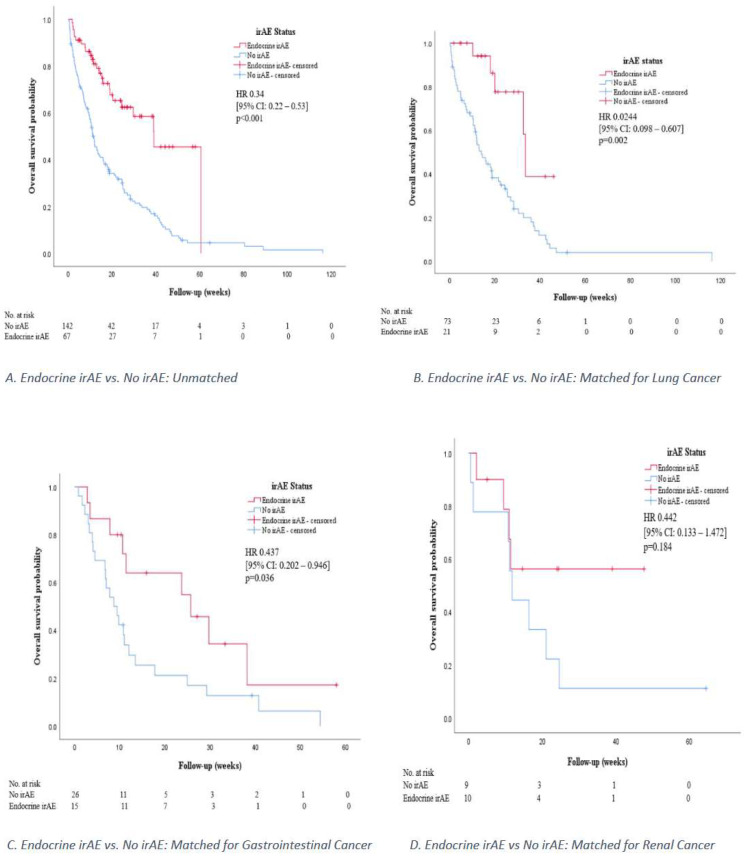
Kaplan–Meier curves for overall survival according to endocrine irAE status.

**Table 1 cancers-14-02687-t001:** Baseline characteristics of study cohort (*n* = 66).

Characteristics	*N*	%
Mean age at start of immunotherapy (years)	60.5 (SD: 10.6)	
Sex		
Male	38	57.6
Primary malignancy		
Head and neck	6	9.1
Lung	22	33.3
Gastrointestinal	15	22.7
Renal	10	15.2
Skin	2	3.0
Gynaecological	8	12.1
Others	3	4.5
Immunotherapy		
Anti-PD-1/Anti-PD-L1	56	84.8
Anti-CTLA-4	2	3.0
Combination	8	12.1
Immune-related endocrine adverse event		
Single endocrinopathy	62	93.9
Thyroid dysfunction	45	72.6
Pituitary dysfunction	15	24.2
Insulin-dependent diabetes mellitus	2	3.2
Multiple endocrinopathies	4	6.1
Median follow-up time (months)	15.7 (range: 1.8–60.5)	

**Table 2 cancers-14-02687-t002:** Characterisation of patients with thyroid dysfunction (*n* = 49).

Characteristics	*N*	%
Pre-existing thyroid conditions		
Present	12	24.5
Median time to onset of thyroid dysfunction (months)	1.8 (range: 0.3–15.8)	
Presenting complaint		
Asymptomatic	33	67.4
Thyrotoxic symptoms	13	26.5
Hypothyroid symptoms	3	6.1
Usage of levothyroxine during thyroid dysfunction		
Yes	35	71.4
Median dosage of levothyroxine during thyroid dysfunction (mcg/kg/day)	1.25 (0.3–3.5)	
Common terminology criteria for adverse events (CTCAE)		
Grade 1 (Mild)	9	18.4
Grade 2 (Moderate)	40	81.6
Grade 3 (Severe)	0	0.0
Grade 4 (Life-threatening)	0	0.0
Grade 5 (Death)	0	0.0

**Table 3 cancers-14-02687-t003:** Patients who developed pituitary dysfunction after immunotherapy (*n* = 18).

Characteristics	*N*	%
Median time to onset of pituitary dysfunction (months)	6.8 (range 1.5–27.3)	
Presenting complaint		
Symptomatic	13	72.2
Type of pituitary dysfunction		
Central hypocortisolism	17	94.4
Multiple pituitary hormone deficiencies	1	5.6
Common terminology criteria for adverse events (CTCAE)		
Grade 1 (Mild)	0	0.0
Grade 2 (Moderate)	15	83.3
Grade 3 (Severe)	3	16.7
Grade 4 (Life-threatening)	0	0.0
Grade 5 (Death)	0	0.0
MRI pituitary/brain		
No pituitary findings	10	55.6
Presence of pituitary lesion	3	16.7
Not done	5	27.8

## Data Availability

The data that support the findings of this study are available from the corresponding author, J.H.L.C., upon reasonable request.

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
