# Peer review of "Clinical Presentation of Immune-Related Endocrine Adverse Events during Immune Checkpoint Inhibitor Treatment"

_cancers, 2022, doi:10.3390/cancers14112687_

Round 1

Reviewer 1 Report

General comments:

The article seeks to examine the association of endocrine immune related adverse events (irAEs) with survival outcomes. The study also aimed to evaluate the clinical course of endocrine irAES and factors associated with persistent dysfunction as well as assess the prognostic significance of endocrine irAEs.

Major comments:

  1. The study is of high interest to scientists and clinicians in the immunotherapy field.
  2. Although the article is well-written, it is descriptive. It will help the readers if they present data in graphical format instead of descriptive so the readers can understand it easily.
  3. Another problem in the study is the small number of patients with a variety of different cancers.
  4. The manuscript lacks other experiments such as basic flow cytometry or relevant serum cytokines in the blood or tissue to gain any biological insight. 
  5. The authors have made a good effort in the discussion section.

Reviewer 2 Report

This is a well-written paper. In the introduction, please consider adding Fujii T et al https://pubmed.ncbi.nlm.nih.gov/29159766/ for irAE and response to tx.

In the discussion, please discuss strategies to mitigate irAE https://pubmed.ncbi.nlm.nih.gov/33310772/.

Reviewer 3 Report

The study demonstrates the immune-related adverse events in immune checkpoint inhibitor treatment affect patient survival.

Figure 3 needs to be revised to accord the labels of irAE Status and No. at risk. Abstract may be revised to indicate abbreviations such as TPO and Tg.

Reviewer 4 Report

Many thanks for this interesting clinical paper.

The authors describe the clinical course of endocrine irAEs (thyroid, pituitary dysfunction, insulin-dependent diabetes mellitus), identify factors associated with persistent thyroid dysfunction, and determine the association between endocrine irAEs and survival parameters in a retrospective single centre study. It includes 66 metastatic oncology patients treated with anti-PD1/anti-CTLA4 and anti-PDL1 for various cancer types .

The english language is fine, and line of reasoning is described well.

I suggest few other points listed below: 

  • line 40: I agree that chemotherapy or targeted therapy show a higher fatal adverse events rate. However, I would state that the profile is  'unique' instead of  ' better' in the sentence: better toxicity profile' of ICIs compared to other systemic anticancer agents. (Adverse effects of immune-checkpoint inhibitors: epidemiology, management and surveillance | Nature Reviews Clinical Oncology)
  • line 43:  Some of the toxicities can happen not during the treatment but long time after the end of the treatment (Delayed immune-related events (DIRE) after discontinuation of immunotherapy: diagnostic hazard of autoimmunity at a distance | Journal for ImmunoTherapy of Cancer | Full Text (biomedcentral.com) Delayed immune-related adverse events with anti-PD-1-based immunotherapy in melanoma - PubMed (nih.gov)
  • line 42: double 'be' ; can occasionally be fatal
  • methods -line 62: did you exclude patients with missing data? 
  • line 87: For the surveillance of irAE instead of 'For surveillance for irAE'
  • line 160: maybe a sentence on the frequence of thyroid irAE and the treatment given? ( or maybe I missed it)
  • line 133: p-value
  • Fig 3:  do you see a OS difference between grade 1 and grade 2 even if your population with thyroid irAEs is highly weighted for grade 2 (40/49)? Is it also significant for the PFS?
  • line 338: ref 40-41 are not in the correct format
  • line 375: Because there are only 3 patients who experienced IDDM, I would be more careful about the conclusion. 
  • discussion: I would state the limitations of this study: low number of patients, single centre and retrospective study. 
  • conclusion: the findings need to be confirmed in larger cohorts. 
